# Relationship of Sporotrichosis and Infected Patients with HIV-AIDS: An Actual Systematic Review

**DOI:** 10.3390/jof9040396

**Published:** 2023-03-23

**Authors:** Rodolfo Pinto-Almazán, Karla A. Sandoval-Navarro, Erika J. Damián-Magaña, Roberto Arenas, Claudia Erika Fuentes-Venado, Paola Berenice Zárate-Segura, Erick Martínez-Herrera, Carmen Rodríguez-Cerdeira

**Affiliations:** 1Sección de Estudios de Posgrado e Investigación, Escuela Superior de Medicina, Instituto Politécnico Nacional, Plan de San Luis y Díaz Mirón, Ciudad de México 11340, Mexico; rodolfopintoalmazan@gmail.com (R.P.-A.); cefvenado@hotmail.com (C.E.F.-V.); pbzars@yahoo.com (P.B.Z.-S.); 2Hospital Central Norte Pemex, Campo Matillas 52, San Antonio, Azcapotzalco, Ciudad de México 02720, Mexico; karlaa.sandoval97@gmail.com; 3Centro Médico Nacional La Raza, Paseo de las Jacarandas S/N, La Raza, Azcapotzalco, Ciudad de México 02990, Mexico; erikadamian2022@gmail.com; 4Sección de Micología, Hospital General “Dr. Manuel Gea González”, Tlalpan, Ciudad de México 14080, Mexico; rarenas98@hotmail.com; 5Efficiency, Quality, and Costs in Health Services Research Group (EFISALUD), Galicia Sur Health Research Institute (IISGS), Servizo Galego de Saúde-Universidade de Vigo (UVIGO), 36213 Vigo, Spain; 6Servicio de Medicina Física y Rehabilitación, Hospital General de Zona No 197, Texcoco 56108, Mexico; 7Dermatology Department, Hospital do Vithas, 36206 Vigo, Spain; 8Fundación Vithas, Grupo Hospitalario Vithas, 28043 Madrid, Spain; 9Department of Health Sciences, University of Vigo, Campus of Vigo, As Lagoas, 36310 Vigo, Spain

**Keywords:** Esporotricosis, VIH-SIDA, *Sporothrix brasiliensis*, *Sporothrix schenckii*, *Sporothrix* spp.

## Abstract

Background: Sporotrichosis is a fungal infection that can affect both humans and animals, caused by a species of thermo-dimorphic fungi of the genus *Sporothrix*. This pathology can be acquired by subcutaneous traumatic inoculation through contact with contaminated plants, soil or decomposing organic matter, and/or by inhalation of conidia. The infection can progress to chronic skin infection, or it can even spread to blood vessels, lymph, muscles, bones, and other organs, such as the lungs and nervous system. Those disseminated types are usually associated with cellular immunodeficiency and infection by inhalation, which explains why people living with human immunodeficiency virus (PLHIV) get infected in such a manner. This virus changes the natural history of sporotrichosis, producing a greater fungal load. Methods: The search was carried out in three databases: Pubmed, Scopus, and Scielo. Eligible articles were considered as those that described sporotrichosis in patients infected with HIV-AIDS, as well as case series. Results: A total of 24 articles were selected, with a sum of 37 patients with sporotrichosis and HIV infection. Of these patients, 31 came from Brazil, two from the United States, one from South Africa, one from Bangladesh, and two from an unspecified region. Regarding epidemiology, a predominance of the male sex was found in 28 of the 37 cases (75.6%), while nine were female (24.3%). Conclusions: Sporotrichosis infection continues to present in a more severe and disseminated way among HIV-positive subjects with lower CD4^+^ counts.

## 1. Introduction

Sporotrichosis is a fungal infection that can affect both humans and animals, caused by a species of thermo-dimorphic fungi of the genus *Sporothrix*. Despite having a worldwide distribution, most reported cases come from tropical and subtropical areas of Latin America, Africa, and Asia [1]. This mycosis has a wide geographical distribution, being mainly found in countries such as the United States, Brazil, Colombia, Guatemala, Mexico, Peru, China, India, Japan, and Australia [2].

Previously, the classification of sporotrichosis related to the Sporothrix schenckii complex, which included: Sporothrix schenckii sensu stricto, Sporothrix globosa, Sporothrix brasiliensis, Sporothrix pallida, Sporothrix luriei, Sporothrix mexicana, and Sporothrix chilensis [3]. However, at present, thanks to scientific advances and with the help of tools such as molecular biology, the term of “species complex” is no longer appropriate; instead, “clinical clade” or “pathogen clade” has been suggested to describe the species most frequently found in humans and animals. Within this clinical clade, we account *S. brasiliensis*, *S. schenckii*, *S. globosa*, and *S. luriei* [4].

However, since 2016, the taxonomic classification of *Sporothrix* spp. changed by clinical clade, including *Sporothrix schenckii*, *S. globosa*, *S. brasiliensis*, and *S. luriei*. On some occasions, species from the environmental clade, such as *S. pallida*, *S. mexicana*, and *S. chilensis*, can cause infection upon contact with the individual [5].

This disease can be acquired by subcutaneous traumatic inoculation through contact with contaminated plants, soil or decomposing organic matter, and/or by inhalation of conidia. Zoonotic transmission mainly caused by felines, such as cats to humans, generally occurs through bites, sneezes and/or scratches, and this is very common in hyperendemic areas [6,7]. There is relevant scientific evidence in which hyperendemic areas of this pathology is emphasized. The most important case is that of Brazil, which suffered from an unprecedented zoonotic outbreak. The geographical expansion of this pathology has been increasing due to the social and health problems in the region, and has spread to new endemic regions. In neighboring countries of Brazil, such as Argentina, Chile, Paraguay, and Uruguay, there are reported cases of Sporotichosis. It should be noted that the virulence factors of the *Sporothrix* genus, such as thermotolerance, melanin synthesis, ergosterol peroxide production, etc., have allowed the increase in infection and pathogen invasion [7,8,9,10,11].

The type of sporotrichosis can be classified into categories by the affected body regions, such as cutaneous and extracutaneous sporotrichosis. The cutaneous comprises three different clinical forms: lymphocutaneous, fixed cutaneous, and disseminated cutaneous [1].

In immunocompetent individuals, most cases are characterized by skin lesions as fixed plaques or nodular lymphangitis. In addition, the infection can progress to a chronic skin infection, or it can disseminate by hematogenous or lymphatic spread to muscles, bones, and other organs, such as the lungs and nervous system [7]. The cutaneous disseminated or systemic forms are usually associated with cellular immunodeficiency, which explains severe sporotrichosis in PLHIV [8]. This virus changes the natural history of sporotrichosis, producing a greater fungal load. Therefore, its opportunistic nature depends on the immune status of the host. Thus, sporotrichosis may be associated with more severe clinical pictures and longer periods of treatment (Figure 1).

Previously, Moreira and collaborators developed a systematic review from 1984 to 2014, reporting 58 cases of patients with HIV coinfection and sporotrichosis [12]. As previously mentioned, this disease has been increasing, becoming a public health problem [13,14]. Because of this, it is of interest to study the existing data on HIV coinfection and sporotrichosis in recent years, which is extremely important. In this systematic review, the epidemiological behavior between both diseases is updated (2015 to 2022), as well as analyzed.

## 2. Materials and Methods

The search was carried out in three databases: Pubmed, Scopus, and Scielo, using the following MESH terms: “sporotrichosis” and “AIDS”; “sporotrichosis” and “HIV”; “*Sporothrix schenckii*” and “AIDS”; “*Sporothrix schenckii*” and “HIV”; “*Sporothrix brasiliensis*” and “HIV”; “*Sporothrix brasiliensis*” and “AIDS”; “*Sporothrix globosa*” and “AIDS”; “*Sporothrix globosa*” ” and “HIV”; “*Sporothrix lurei*” and “AIDS”; “*Sporothrix lurei*” and “HIV”; “*Sporothrix pallida*” and “AIDS”; “*Sporothrix pallida*” and “HIV”; “*Sporothrix schenckii sensu lato*” and “AIDS”; “*Sporothrix schenckii sensu lato*” and “HIV”; from the year 2015 (after the systematic review carried out by Moreira and colleagues) until 2022, which gave a total of 104 results. The search was limited to studies in humans and performed without any language restrictions. In order to carry out the systematic review, we used the article guidelines for systematic reviews and meta-analysis PRISMA 2020 (Figure 2).

Eligible articles were considered those that described sporotrichosis in patients infected with HIV-AIDS, as well as case series. Articles that were reviews from literature, retrospective studies outside the proposed limit, cohort studies, studies of patients with sporotrichosis, but who did not have HIV-AIDS, studies of patients with HIV-AIDS without sporotrichosis, and book chapters were excluded. The following data were extracted: age, sex, sporotrichosis species, technique for identification of the species, mode of transmission, topography, clinical presentation, temporality, treatment, clinical evolution of the patient, CD4^+^ count, culture area, and MESH term (Figure 2).

## 3. Results

A total of 25 articles were selected, with a sum of 52 patients with sporotrichosis and HIV infection [15,16,17,18,19,20,21,22,23,24,25,26,27,28,29,30,31,32,33,34,35].

### 3.1. Epidemiology

Of these patients, 46 came from Brazil, two from the United States, one from South Africa, one from Bangladesh, and two from an unspecified region. Regarding epidemiology, a predominance of the male sex was found in 41 of the 52 cases (78.85%), while 11 were female (21.15%) [15,16,17,18,19,20,21,22,23,24,25,26,27,28,29,30,31,32,33,34,35].

Interestingly, even when Moreira et al. covered a longer period of time (20 years), 58 patients studied were reported (2.9 cases per year), whilst in the present work, 52 patients were found across 8 years (6.5 patients per year) [12]. The age range was from 20 to 59 years, with a mean of 36 years. Regarding the average age, Moreira et al. reported 37.8 ± 10.4 years [9].

Likewise, as reported by Moreira et al., the largest number of cases were reported in Brazil, with a predominance of males [15,16,17,18,19,20,21,22,23,24,25,26,27,28,29,30,31,32,33,34]. This could be interpreted in two ways: firstly, that the largest number of HIV cases are from males, as reported by the epidemiological services of Brazil, and other affected countries, such as the US and South Africa [36,37,38]; and secondly, as mentioned above, that Brazil is a hyperendemic country for sporotrichosis due to the extensive zoonotic transmission in that area (cats) [9,10].

### 3.2. Sporotrichosis and Topography

Even though culture and histopathology continue to be the most recurrent ways to identify *Sporothrix* spp., there are molecular biology techniques, such as simple PCR, real-time PCR, etc., which enable a more precise identification of the etiology of the fungus, using both genes and molecular markers [5]. For example: beta-tubulin and calmodulin have been found to be the most resolute for the identification of the species of this genus of fungus, which allowed us to observe that the causative agent with the highest frequency was *S. brasiliensis*, with 67.31% (35/52), followed by *S. schenckii*, with 9.61% (5/52). However, despite the existence of molecular techniques, seven patients with *Sporothrix* spp. isolates were reported in some articles of this systematic review, representing 13.46% (7/52), and in five patients, the etiologic agent was not isolated, representing 9.62% (5/52) [15,16,17,18,19,20,21,22,23,24,25,26,27,28,29,30,31,32,33,34,35]. As the study by Moreira et al. was released prior to the new classification by clades, and fungal culture was the only method used for identification, *Sporothrix schenckii* was mentioned as the only causative agent (Table 1) [12].

The most frequently observed clinical varieties were the disseminated extracutaneous (32/52), disseminated cutaneous (8/52), and disseminated (8/52), which correspond to 89.19% of the cases. In fourth place, the localized cutaneous variety was identified (2/52) in 5.41% of the patients and finally, the lymphocutaneous and cutaneous generalized varieties (1/52; 1.92%) [15,16,17,18,19,20,21,22,23,24,25,26,27,28,29,30,31,32,33,34,35]. Concerning the prevalence and in discrepancy with the present study, Hernández-Castro et al. previously reported that in Latin America (hyperendemic region), the cutaneous lymphangitic form is one that occurs most frequently, followed by the fixed cutaneous. It should be noted that this study was conducted including all cases of sporotrichosis, without considering whether or not they were affected by any comorbidity [5].

Regarding topography, the most affected areas were the head and neck (32%), upper limbs (21.6%), and lower limbs (18.9%). On the other hand, extracutaneous disseminated (ECD) sporotrichoses, also known as disseminated systemic, are severe forms and therefore rare types, however, they were the cases with the highest prevalence within this study (*n* = 32). Additionally, even though there were 32 cases, these patients were affected in a multisystem or organic way, which included the respiratory system (upper and lower airways) (*n* = 10), sense of sight (*n* = 9), skeletal system (*n* = 9), and central nervous system (CNS) (*n* = 21) [15,16,17,18,19,20,21,22,23,24,25,26,27,28,29,30,31,32,33,34,35]. Nevertheless, this can be understood, as mucous membranes can be affected after hematogenous dissemination, which is very common in patients with immunosuppression caused by HIV [39]. It is important to mention that even when CNS ECDs are recurrent cases in PLHIV, in several of the reported cases, it was impossible to isolate the pathogen, or in its case, to perform molecular identification of the type of sporotrichosis due to the low burden that the patients had [21,35].

Similarly, Moreira et al. did not report the affected topography of each patient. However, when searching for the articles included in his study, it was found that the head, neck, and upper limbs were the most affected sites, with 20.6% (12 patients), followed by the lower limbs, with 17.2% (10 patients). Those topographical findings are consistent with the ones from our review, demonstrating that sporotrichosis continues to affect body segments in the same proportion over time (Table 1) [12].

Additionally, regarding ECD types, sporotrichosis in PLHIV can be aggravated because subjects can generate the immune reconstitution inflammatory syndrome (IRIS). IRIS is classified as the manifestation and/or worsening of pre-existing sporotrichosis by immune recovery, when the CD4^+^ T-cell count normalizes or HIV viral load decreases. This dysregulated immune response to sporotrichosis can be generated after the institution of antiretroviral treatment, as well as low adherence to them [35]. During the period of this study, there was only one original article in which 15 cases of IRIS sporotrichosis meningitis were reported. According to this report, this form of sporotrichosis has high mortality (10/15 patients) [35].

### 3.3. Modes of Transmission and Diagnosis

Regarding the mode of transmission of sporotrichosis, 33 of the 52 patients had a history of contact with cats as a form of transmission: four of them reported scratches and three bites. In addition, an environmental history was found in seven of the 52 patients, of which three reported contact with plants, one had contact with contaminated water, one suffered trauma with glass, one patient had a cut with a sharp device, and one informed about contact with the surrounding environment. Finally, nine cases did not provide an explanation of the source of infection [15,16,17,18,19,20,21,22,23,24,25,26,27,28,29,30,31,32,33,34,35]. An important difference between both systematic reviews is that in the one carried out by Moreira et al., the most frequent form of sporotrichosis transmission is not reported. When analyzing their systematic review, it was found that only two (trauma caused by flower thorns) of the 58 cases (3.4%) informed their transmission mechanism. Fifty-two of the 58 patients (89.6%) included in their study did not mention this information, and, finally, in four of the 58 patients (6.8%), the data could not be obtained because they did not have access to the articles. This analysis allows us to understand that while trauma by flora was previously reported as the main mode of transmission, currently, other causes such as animal bites or scratches are reported, with the domestic cat being the main disease vector (Table 1) [12].

Concerning the diagnosis, it has been stated that the fungal culture continues to be the gold standard for the identification of the genus of the fungus (52/52), followed by histopathology (2/52). Some authors identified species of *S. schenckii* through culture in their cases. This happened because previously, the different species included in the *S. schenckii* complex, described before the new clade nomenclature, were not known. However, with the use of molecular biology techniques, such as simple PCR, amplifying with different genes, and molecular markers (beta-tubulin and calmodulin), it was possible to identify the species involved in most case reports (35/52), and in one case, the method used was not specified. Cerebrospinal fluid (30.77%), skin biopsy (25.00%), and sputum (23.08%) were the main samples evaluated (Table 1) [15,16,17,18,19,20,21,22,23,24,25,26,27,28,29,30,31,32,33,34,35].

### 3.4. Treatment and Evolution of the Patient

In regard to the antifungal regimens found, the most frequently used was Amphotericin B in combination with Itraconazole in 23.08% of the patients (12/52), followed by monotherapy with Amphotericin B in 28.85% (15/52) (Table 1) [15,16,17,18,19,20,21,22,23,24,25,26,27,28,29,30,31,32,33,34,35]. On the other hand, and similar to our observations, in the review by Moreira et al., the election treatment was Amphotericin B in combination with Azoles in 20 patients (34.5%), followed by Azole monotherapy in 19 patients (32.8%) [12]. These comparable schemes may imply that pharmacological management has prevailed throughout the years. Though, it should be clarified that the chronic use of Amphotericin B deoxycholate and lipid formulations can cause important nephrotoxic effects, and due to this, they are recommended mainly for the treatment of both pulmonary and meningeal sporotrichosis, and for disseminated cutaneous highly resistant to other antifungal treatments. They can also be used in cases of osteoarticular sporotrichosis, where the intraarticular region is not involved. One of the great advantages of this family of drugs is that its possible to use during pregnancy. Another treatment that has been effective for cutaneous (fixed and lymphocutaneous) and extracutaneous (osteoarticular) forms is itraconazole; however, this drug has long-term toxic effects on the stratum corneum, causes nausea, edema, epigastrium pain, hypercholesterolemia/hypertriglyceridemia, as well as impairs liver function [39,40,41,42].

As for the time of evolution of the disease, a range of 2 weeks to 9 years was found, with an average of 20 months. Equally, the mean CD4^+^ cell count was 104 cells/mm^3^ (Table 1) [15,16,17,18,19,20,21,22,23,24,25,26,27,28,29,30,31,32,33,34,35]. Temporality is another variable that Moreira et al. did not report. This may be because in most of the case reports within their systematic review, they did not mention it (68.4%). Opposite to our findings, the duration of the disease reported the most was for less than 1 year (8%). Nevertheless, we do not have enough information to compare the previous temporality with the current one, so we cannot make any further conclusions about that matter [12].

Finally, regarding the outcome, 24 of the 52 patients were reported with favorable evolution (46.15%), 22 patients died (42.30%), and six patients were lost to follow-up (16.2%), so their evolution is unknown (Table 1) [15,16,17,18,19,20,21,22,23,24,25,26,27,28,29,30,31,32,33,34,35]. It is worth mentioning that even though some patients were reported with favorable evolution and others were lost before concluding their treatment, it does not mean that they could not fall back in the infection, because the follow-up of sporotrichosis must be long-term, as it is a chronic infection with many relapses, especially in PLHIV. In agreement with the review by Moreira et al., we also informed that the majority of patients survived (70%) [12].

With the above, we can infer that despite the fact that both diseases occur in a context where the patient is very vulnerable in relation to their own health, a proper management may allow for a hopeful resolution.

## 4. Conclusions

The prevalence of sporotrichosis coinfection per year (6.5 cases) has increased in the last 8 years compared with what was previously observed by Moreira et al. This is largely due to the fact that sporotrichosis is an emerging disease, especially in hyperendemic regions of the world, which has spread from zoonotic contamination by felines.

Sporotrichosis disease continues to occur in a more severe and widespread manner among HIV-positive subjects with lower CD4+ counts. However, paradoxically, there are cases with previous sporotrichosis that worsen at the time of initiation of treatment with retrovirals in PLHIV, causing IRIS. Both IRIS and ECD sporotrichosis are the rarest and most aggressive types of sporotrichosis, and therefore have high incidence and lethality in PLHIV.

Likewise, it must be emphasized that the HIV test should be performed routinely in patients with acquired sporotrichosis in endemic areas, to improve the prognosis and quality of life of patients with the appropriate treatment, and avoid IRIS.

It should be noted that with the advance of molecular diagnostic techniques, it has been possible to analyze the causative pathological agent of sporotrichosis in these patients more precisely. Nevertheless, there are cases in which it is complicated to take enough sample to make the diagnosis of *Sporothrix* species, as in the case of meningitis.

The investigation of the association of the sporotrichosis and HIV is crucial for the generation of comprehensive measures for the benefit of patients.

## Figures and Tables

**Figure 1 jof-09-00396-f001:**
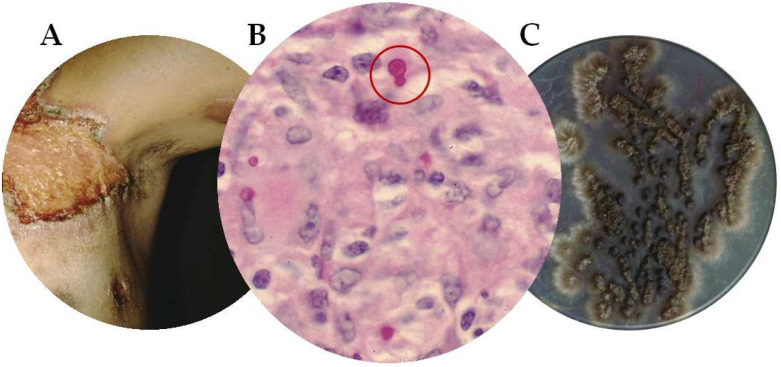
Common characteristics of Sporotrichosis. (**A**) *Sporothrix* chancre, (**B**) *Sporothrix* yeasts on PAS-stained smears, (**C**) culture of *Sporothrix* spp. on Sabouraud dextrose agar with folded appearance.

**Figure 2 jof-09-00396-f002:**
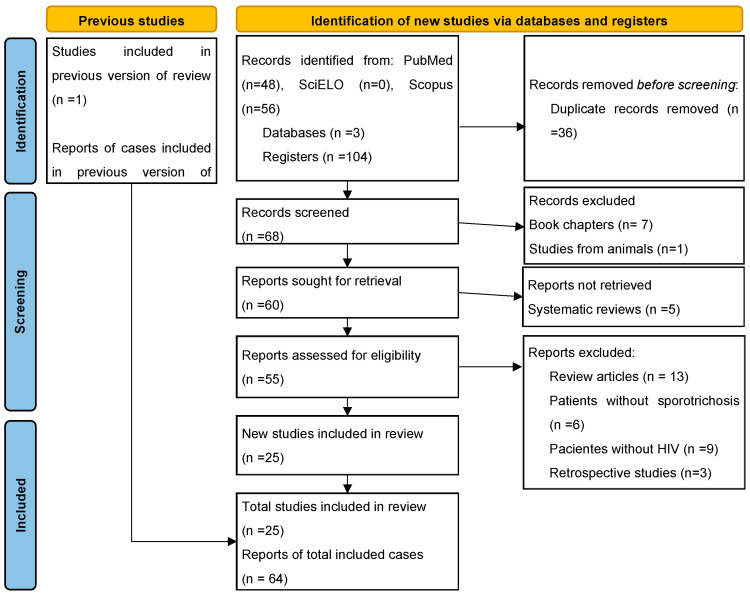
PRISMA flow diagram of data extracted from bibliographic searches.

**Table 1 jof-09-00396-t001:** Cases of Sporotrichosis in patients with HIV (2015–2022).

Case	Country	Age (Years)/ Sex	Sporotrichosis Type	Topography/ Sample Type	Mode of Transmission	Sporothrix Specie	Identification Method	Treatment	Evolution/Outcome	CD4/mm^3^	Ref.
1	Brazil	31/M	ECD	Skin and eyes (choroid)/skin biopsy	Domestic cat bite	*S. schenckii*	SDA	AmB	2 weeks/ favorable	9	[15]
2	Brazil	45/M	D	NA/skin biopsy	Contact with domestic cats	*S. brasiliensis*	Endpoint PCR	AmB	1 year/death	46	[16]
3		37/M	D	NA/blood	Domestic cat bite	*S. brasiliensis*	Endpoint PCR	AmB	1 year/Tx	66	
4		38/M	D	NA/skin biopsy	Contact with domestic cats	*S. brasiliensis*	Endpoint PCR	AmB	2 years/death	53	
5		25/M	D	NA/bone marrow	NA	*S. brasiliensis*	Endpoint PCR	AmB	3 years/death	25	
6		20/F	D	NA/skin biopsy	Contact with domestic cats	*S. brasiliensis*	Endpoint PCR	AmB	1 year/death	111	
7	Brazil	35/M	ECD	Ophtalmic/blood, bone marrow, lymph nodes	Contact with domestic cats	*S. brasiliensis*	Endpoint PCR	AmB + ITZ	7 years/ favorable	53	[17]
8		25/M	ECD	Ophtalmic/blood, laryngeal mucus, pharynx	Direct trauma with pointed device	*S. brasiliensis*	Endpoint PCR	AmB + ITZ	8 years/NA	25	
9		43/M	ECD	Ophtalmic/blood, nasal mucus, sputum	Contact with domestic cats	*S. brasiliensis*	Endpoint PCR	AmB + TRF	9 years/NA	35	
10	Brazil	47/F	CD	Face, upper and lower limbs/cutaneous scales	NA	*Sporothrix* spp.	SDA	AmB + ITZ	3 months/death	47	[18]
11	Brazil	38/F	CD	Generalized/skin biopsy	Contact with domestic cats	*S. brasiliensis*	SDA and endpoint PCR	AmB	2.5 months/favorable	62	[19]
12	USA	40/M	C	Posterior part from right leg/skin biopsy	NA	*S. schenckii*	Histopathology	ITZ	1 year/favorable	140	[20]
13	Brazil	20/F	D	Face, truck and upper limbs/blood, CSF, sputum and urine	Domestic cat scratch	*S. brasiliensis*	SDA and endpoint PCR	AmB + ITZ + TRF	2 years/death	111	[21]
14	Brazil	40/F	CD	Oral cavity, nasal septum and maxilar sinus/oral ulcers biopsy	Contact with domestic cats	*S. brasiliensis*	SDA and endpoint PCR	AmB + ITZ	1 year/favorable	5	[22]
15	USA	30/M	ECD	CNS and left hand/cerebral tissue biopsy	NA	*S. schenckii*	SDA	AmB + ITZ	>6 weeks/favorable	64	[23]
16	Brazil	52/M	ECD	Skin, lungs and bone/first morning sputum	Contact with plants and products from wood	*S. brasiliensis*	SDA, LJ medium and endpoint PCR	AmB + ITZ	NA/favorable	46	[24]
17		25/M	ECD	Skin, lungs and bone/first morning sputum	Contact with plants and products from wood	*S. brasiliensis*	SDA, LJ medium and endpoint PCR	AmB + ITZ + PSZ + TRF	NA/death	25	
18		31/F	ECD	Skin, lungs and bone/first morning sputum	Contact with domestic cats	*S. brasiliensis*	SDA, LJ medium and endpoint PCR	AmB + ITZ	NA/favorable	21	
19		20/F	ECD	Skin, bone and CNS/first morning sputum	Domestic cat scratch	*S. brasiliensis*	SDA, LJ medium and endpoint PCR	AmB + ITZ + PSZ + TRF	NA/death	42	
20		44/M	ECD	Skin, bone and CNS/first morning sputum	Contact with plants and products from wood	*S. brasiliensis*	SDA, LJ medium and endpoint PCR	AmB + ITZ	NA/death	110	
21		26/M	ECD	Skin, CNS, upper airway/first morning sputum	Contact with domestic cats	*S. brasiliensis*	SDA, LJ medium and endpoint PCR	AmB + ITZ	NA/death	178	
22		36/M	ECD	Skin, CNS, upper airway/first morning sputum	Domestic cat scratch	*S. brasiliensis*	SDA and endpoint PCR	AmB + TRF + ITZ	Loss of follow-up	66	
23		46/M	ECD	Skin, bone, upper airway, eyes/first morning sputum	Domestic cat bite	*S. brasiliensis*	SDA and endpoint PCR	ITZ + AmB	NA/cured	35	
24		35/M	ECD	CNS, eyes, skin, bone, upper airway/first morning sputum	Contact with domestic cats	*S. brasiliensis*	SDA and endpoint PCR	ITZ + AmB	NA/death	75	
25		43/M	ECD	Skin, bone, upper airway, eyes/first morning sputum	Domestic cat scratch	*S. brasiliensis*	SDA and endpoint PCR	ITZ + AmB + PSZ	NA/death	35	
26		20/F	ECD	Skin, bone, upper airway, eyes/first morning sputum	Domestic cat scratch	*S. brasiliensis*	SDA and endpoint PCR	ITZ + PSZ + AmB	NA/cured	56	
27	South Africa	NA/NA	CD	Face, upper and lower limbs/skin biopsy	NA	*S. schenckii*	SDA	AmB	NA/NA	59	[25]
28	Brazil	59/M	C	Left hand/skin biopsy	Contaminated water	*S. schenckii*	SDA	ITZ	14 months/favorable	558	[26]
29	Brazil	47/M	D	Face, hands, oral cavity, legs/skin biopsy	NA	*S. brasiliensis*	SDA and endpoint PCR	NA	NA/NA	47	[27]
30	NA	23/M	CD	Upper and lower limbs/skin biopsy	NA	*Sporothrix* spp.	SDA	ITZ + AmBd, PDN	NA/favorable	43	[28]
31		37/M	CD	Upper limbs, neck and trunk/skin biopsy	NA	*Sporothrix* spp.	SDA	ITZ + AmBd, PDN	NA/favorable	66	
32	Brazil	34/M	CD	Generalized/skin biopsy	Contact with domestic cats	*Sporothrix* spp.	SDA	AmBd, TMP-SMX and Azitromicin	2 months/favorable	6	[29]
33	Brazil	38/M	CG	Face, feet, hands, glans penis/NA	Contact with domestic cats	*Sporothrix* spp.	Histopathology	AmBd, ITZ	10 months/ favorable	249	[30]
34	Brazil	52/M	LC	Back of left foot/foot biopsy	Trauma with glass	*S. brasiliensis*	SDA and endpoint PCR	ITZ + cryosurgery	2.25 years/favorable	809	[31]
35	Brazil	38/M	D	NA	NA	*Sporothrix* spp.	NA	AmB + PSZ	5 months/ favorable	106	[32]
36	Bangladesh	35/F con 30 SDG	ECD	Eye/palpebral conjunctiva	Contact with a cat contaminated with sporotrichosis	*S. schenckii*	SDA and endpoint PCR	ARV + Intralesional AmB (5 mg only dose) + FCZ in each eye every 4 h	NA/favorable	221	[33]
37	South Africa	NA/M	CD	Head, neck, upper and lower limbs/pus	Contact with surrounding environment	*Sporothrix* spp.	SDA and endpoint PCR	VCZ	NA/favorable	NA	[34]
38	Brazil	44/M	ECD	CNS/CSF	12 cases out of 17 had close contact with cats	*S. brasiliensis*	SDA, Mycosel agar and PCR	AmB + PSZ	15 days/death	median CD4+ T lymphocyte count was 110 cells/mm^3^ (range 8–704 cells/mm3)	[35]
39		57/M	ECD	CNS/CSF		NO	SDA, Mycosel agar and PCR	AmB + PSZ	1 month/favorable		[35]
40		28/M	ECD	CNS/CSF		NO	SDA, Mycosel agar and PCR	AmB	2 month/favorable		[35]
41		46/M	ECD	CNS/CSF		NO	SDA, Mycosel agar and PCR	AmB	4 month/death		[35]
42		26/M	ECD	CNS/CSF		*S. brasiliensis*	SDA, Mycosel agar and PCR	AmB	15 days/death		[35]
43		37/M	ECD	CNS/CSF		NO	SDA, Mycosel agar and PCR	ITZ	1 month/death		[35]
44		48/M	ECD	CNS/CSF		*S. brasiliensis*	SDA, Mycosel agar and PCR	AmB	15 days/death		[35]
45		21/F	ECD	CNS/CSF		*S. brasiliensis*	SDA, Mycosel agar and PCR	AmB + PSZ	1 month/death		[35]
46		40/M	ECD	CNS/CSF		NO	SDA, Mycosel agar and PCR	AmB	2 month/death		[35]
47		31/F	ECD	CNS/CSF		*S. brasiliensis*	SDA, Mycosel agar and PCR	AmB + PSZ	1 month/death		[35]
48		38/M	ECD	CNS/CSF		*S. brasiliensis*	SDA, Mycosel agar and PCR	AmB	1 month/death		[35]
49		43/M	ECD	CNS/CSF		*S. brasiliensis*	SDA, Mycosel agar and PCR	AmB	8 month/favorable		[35]
50		23/M	ECD	CNS/CSF		*S. brasiliensis*	SDA, Mycosel agar and PCR	AmB + PSZ	15 month/favorable		[35]
51		35/M	ECD	CNS/CSF		*S. brasiliensis*	SDA, Mycosel agar and PCR	AmB	2 month/death		[35]
52		20/M	ECD	CNS/CSF		*S. brasiliensis*	SDA, Mycosel agar and PCR	AmB + PSZ	3 month/favorable		[35]

M = masculine; F= feminine; ECD = extracutaneous disseminated; D = disseminated; C = cutaneous; CD = cutaneous disseminated; CG = cutaneous generalized; LC = lymphocutaneous; SDA = Sabouraud dextrose agar; PCR = polymerase chain reaction; NA = not available; Tx = treatment; CSF = cerebrospinal fluid; CNS = central nervous system; WG = weeks of gestation; ARV = antiretroviral; AmB = amphotericin B; AmBd = amphotericin B deoxycholate; ITZ = Itraconazole; PSZ = Posaconazole; FCZ = Fluconazole; VCZ = Voriconazole; TRF = Terbinafina; TMP-SMX = Trimetropin sulfamethoxazole; LJ = Löwenstein–Jensen.

## Data Availability

Not applicable.

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
