# Peer review of "Relationship of Sporotrichosis and Infected Patients with HIV-AIDS: An Actual Systematic Review"

_jof, 2023, doi:10.3390/jof9040396_

Round 1

Reviewer 1 Report

This article is an actual systematic review of relationship of sporotrichosis and infected patients with HIV. The authors update the study by Moreira and collaborators who developed a systematic review from 1984 to 2014.

The search was carried out in three databases; Pubmed, 35 Scopus and Scielo. Eligible articles were considered as those that described sporotrichosis in patients infected with HIV-AIDS, as well as case series. A total of 24 articles were selected, 37 with a sum of 37 patients with sporotrichosis and HIV infection. Of these patients, 31 came from 38 Brazil, 2 from the United States, 1 from South Africa, 1 from Bangladesh, and 2 from an unspecified 39 region. Regarding epidemiology, a predominance of male sex was found in 28 of the 37 cases 40 (75.6%), while 9 were female (24.3%). The authors concluded that sporotrichosis infection continues to present in a more severe and disseminated way among HIV-positive subjects with lower CD4+ counts.

Reviewer :The subject is very important but some points I would like to point:

 I- Introduction

a. “This pathology can be acquired by subcutaneous traumatic inoculation through contact with contaminated plants, soil or decomposing organic matter, and/or by inhalation of conidia. Zoonotic transmission mainly caused by felines such as cats to humans, generally occurs through bites, sneezes and/or scratches, and this is very common in hyperendemic areas”.

Reviewer:

Considering the many of cohort studies that were published about hyperendemic areas in Brazil, this paragraph need more references to be contextualized.The authors just write the reference of Gremião et al, 2021, that is a " Guideline for the Management of Feline Sporotrichosis Caused by Sporothrix Brasiliensis and Literature Revision".

 b. “The type of sporotrichosis can be classified into categories by the affected body regions, such as cutaneous and extracutaneous sporotrichosis. This last one comprises four different clinical forms: lymphocutaneous, fixed cutaneous, disseminated cutaneous, and extracutaneous [1]”.

 Reviewer:

It is not corrected. Lymphocutaneous sporotrichosis, fixed cutaneous and disseminated cutaneous are cutaneous forms and not are extracutaneous forms. 

c. “In immunocompetent individuals, most cases are characterized by presenting skin  lesions in the form of fixed plaques or affecting the lymph vessels, causing the nodular ascending lymphangitic form. In addition, the infection can progress to a chronic skin infection, or it can even spread to lymph vessels, muscles, bones, and other organs such as  the lungs and nervous system [7]. These disseminated types are usually associated with cellular immunodeficiency and inhalation infection, which explains why the population with the acquired immunodeficiency virus (HIV) gets contaminated [8]. These disseminated types are usually associated with cellular immunodeficiency and inhalation infection, which explains why the population  with the acquired immunodeficiency virus (HIV) gets contaminated [8]”. 

Reviewer:

-The authors did not mention the hematogenous route that is very important in the dissemination of the fungus.

 -Typically, when the fungus is acquired by direct inhalation of conidia, patients have underlying respiratory conditions (mainly chronic obstructive pulmonary disease) and develop primary pulmonary sporotrichosis. More commonly, the lung is affected in immunosuppressed patients, by hematogenous or lymphatic spread from a distal site due to immunosuppression. The inhalation route is not common in immunosuppressed patients. 

-Population with the acquired immunodeficiency virus (HIV). The better term is People living with HIV (PLHIV).

II- Results and discussion:

 a. Reviewer:

Most cases are from Brazil. Epidemiological, diagnostic, clinical and treatment aspects have already been published. The authors do not discuss or deepen very relevant aspects of the HIV/sporotrichosis association, which are the disseminated forms, mainly in the central nervous system and immune inflammatory syndrome of these patients when starting antiretroviral therapy. They just compare their results with Moreira et al.

 b. In the text:  ”However, it must be considered that advances in medical research have evolved exponentially in recent years, creating new drugs, as well as therapeutic regimens, increasing the number of cured patients, as well as a longer life expectancy. This might be a plausible explanation for the longer average duration of the disease in our report, since, with better HIV management, patients have greater survival that ultimately means a longer life 240 with comorbidities (like sporotrichosis).”

 Reviewer:

I do not agree with this paragraph. Considering in Results: “Finally, speaking of the outcome, 19 of the 37 patients reported favorable evolution 232 (59.3%), 12 patients died (32.4%) and 6 patients were lost to follow-up (16.2%), so their 233 evolution is unknown (Table 1) [12–31]”.

There was a high number of patients that died (32.4%) and six patients were lost to follow-up (16.2%). The follow-up of sporotrichosis must be long because is a chronic infection with many relapses meanly in PLHIV.

Amphotericin B is the only intravenous antifungal available to treat severe sporotrchosis and was introduced for clinical use in the 1960s. Although lipid formulations were incorporated into the therapeutic arsenal, they can still be potentially toxic and often require treatment interruption. In addition, AMB´s in vitro studies demonstrates moderate susceptibility of Sporothrix. 

III-Conclusions:

There are no new aspects to the subject in the conclusions to those already  published articles.

Figure 1:

The quality of the Figure 1 A and C are not good.

Author Response

Answers to Reviewer 1 concerns:

This article is an actual systematic review of relationship of sporotrichosis and infected patients with HIV. The authors update the study by Moreira and collaborators who developed a systematic review from 1984 to 2014.

The search was carried out in three databases; Pubmed, 35 Scopus and Scielo. Eligible articles were considered as those that described sporotrichosis in patients infected with HIV-AIDS, as well as case series. A total of 24 articles were selected, 37 with a sum of 37 patients with sporotrichosis and HIV infection. Of these patients, 31 came from 38 Brazil, 2 from the United States, 1 from South Africa, 1 from Bangladesh, and 2 from an unspecified 39 region. Regarding epidemiology, a predominance of male sex was found in 28 of the 37 cases 40 (75.6%), while 9 were female (24.3%). The authors concluded that sporotrichosis infection continues to present in a more severe and disseminated way among HIV-positive subjects with lower CD4+ counts.

Reviewer: The subject is very important but some points I would like to point:

We appreciate the time and effort you have invested in the revision of our manuscript. Indeed, all your suggestions have improved the quality of our manuscript. In the main text, the additions are highlighted in yellow. We hope that we have correctly addressed all your concerns.

  1. I- Introduction
  2. “This pathology can be acquired by subcutaneous traumatic inoculation through contact with contaminated plants, soil or decomposing organic matter, and/or by inhalation of conidia. Zoonotic transmission mainly caused by felines such as cats to humans, generally occurs through bites, sneezes and/or scratches, and this is very common in hyperendemic areas”.

Reviewer:

Considering the many of cohort studies that were published about hyperendemic areas in Brazil, this paragraph need more references to be contextualized. The authors just write the reference of Gremião et al, 2021, that is a " Guideline for the Management of Feline Sporotrichosis Caused by Sporothrix brasiliensis and Literature Revision".

Answer: Thank you for your valuable observation. The pertinent modifications were made, which are highlighted in yellow in the paper.

Lines 68-76: “There is relevant scientific evidence in which they emphasize hyperendemic areas of this pathology. The most important case is that of Brazil, where they have suffered an unprecedented zoonotic outbreak. The geographical expansion of this pathology has been increasing due to the social and health problems in the region and has spread to new endemic regions. In neighboring countries of Brazil such as Argentina, Chile, Paraguay and Uruguay there are reported cases of Sporotichosis. It should be noted that the virulence factors of the Sporothrix genus such as thermotolerance, melanin synthesis, ergosterol peroxide production, etc. have allowed the increase in infection and pathogen invasion [7–11].”

  1. The type of sporotrichosis can be classified into categories by the affected body regions, such as cutaneous and extracutaneous sporotrichosis. This last one comprises four different       clinical forms: lymphocutaneous, fixed cutaneous, disseminated cutaneous, and           extracutaneous [1]”

            Reviewer: It is not corrected. Lymphocutaneous sporotrichosis, fixed cutaneous and     disseminated cutaneous are cutaneous forms and not are extracutaneous forms.

Answer: Thank you for your valuable input. We had added your suggestion.

Lines 77-80: “The type of sporotrichosis can be classified into categories by the affected body regions, such as cutaneous and extracutaneous sporotrichosis. The cutaneous comprises three different Clinical forms: lymphocutaneous, fixed cutaneous and disseminated cutaneous [1].”

  1. “In immunocompetent individuals, most cases are characterized by presenting skin lesions in the form of fixed plaques or affecting the lymph vessels, causing the nodular ascending lymphangitic form. In addition, the infection can progress to a chronic skin infection, or it can even spread to lymph vessels, muscles, bones, and other organs such as the lungs and nervous system [7]. These disseminated types are usually associated with cellular immunodeficiency and inhalation infection, which explains why the population with the acquired immunodeficiency virus (HIV) gets contaminated [8]. These disseminated types are usually associated with cellular immunodeficiency and inhalation infection, which explains why the population with the acquired immunodeficiency virus (HIV) gets contaminated [8]”.

Reviewer:

-The authors did not mention the hematogenous route that is very important in the dissemination of the fungus.

-Typically, when the fungus is acquired by direct inhalation of conidia, patients have underlying respiratory conditions (mainly chronic obstructive pulmonary disease) and develop primary pulmonary sporotrichosis. More commonly, the lung is affected in immunosuppressed patients, by hematogenous or lymphatic spread from a distal site due to immunosuppression. The inhalation route is not common in immunosuppressed patients. 

Answer: Thank you for your valuable input. We had added your suggestion.

Lines: 81-87:” In immunocompetent individuals, most cases are characterized by skin lesions as fixed plaques or nodular lymphangitis. In addition, the infection can progress to a chronic skin infection, or it can disseminate by hematogenous or lymphatic spread to muscles, bones, and other organs such as the lungs and nervous system [7]. The cutaneous disseminated or systemic forms are usually associated with cellular immunodeficiency, which explains severe sporotrichosis in people living with human immunodeficiency virus (PLHIV) patients [8].”

  1. -Population with the acquired immunodeficiency virus (HIV). The better term is People living with HIV (PLHIV).

Answer: Thank you for your valuable input. We had added your suggestion.

The observation was addressed on the following lines 33-34 y 86.

  1. II- Results and discussion:

  1. Reviewer:

Most cases are from Brazil. Epidemiological, diagnostic, clinical and treatment aspects have already been published. The authors do not discuss or deepen very relevant aspects of the HIV/sporotrichosis association, which are the disseminated forms, mainly in the central nervous system and immune inflammatory syndrome of these patients when starting antiretroviral therapy. They just compare their results with Moreira et al.

Answer: Thank you for your valuable input. We had added your suggestion.

Lines 192 -197: Concerning the prevalence and in discrepancy with the present study, Hernández-Castro et al. previously reported that in Latin America (hyperendemic region) the cutaneous lymphangitic form is the one that occurs most frequently, followed by the fixed cutaneous. It should be noted that this study was conducted of all cases of sporotrichosis without considering whether or not they suffered from any comorbidity. [5]

Lines 199 – 210: “On the other hand, extracutaneous disseminated (ECD) sporotrichoses or also known as disseminated systemic are severe forms and therefore rare types, however, they were the cases with the highest prevalence within this study (N = 32). Also, even though there were 32 cases, these patients were affected in a multisystem or organic way, which included the respiratory system (upper and lower airways) (N = 10), sense of sight (N = 9), skeletal system (N = 9) and central nervous system (CNS) (N = 21) [15–35]. Nevertheless, this can be understood, as mucous membranes can be affected after hematogenous dissemination, which is very common in patients with immunosuppression caused by HIV [39]. It is important to mention that even when CNS ECDs are recurrent cases in PLHIV, in several of the reported cases it was impossible to isolate the pathogen, or in its case to perform molecular identification of the type of sporotrichosis due to the low burden that patients have [21,35].”

Lines 217 -225: “Additionally, to the ECD types, sporotrichosis in PLHIV can be aggravated because subjects can generate the immune reconstitution inflammatory syndrome (IRIS). IRIS is classified as the manifestation and/or worsening of pre-existing of sporotrichosis by immune recovery when CD4+ T cell count normalizes or HIV viral load decreases. This dysregulated immune response to sporotrichosis can be generated after the institution of antiretroviral treatment, as well as low adherence to them [35]. During the period of this study, there is only one original article in which 15 cases of IRIS sporotrichosis meningitis were reported. According to this report, this form of sporotrichosis has high mortality (10 / 15 patients) [35]

  1. In the text:” However, it must be considered that advances in medical research have evolved exponentially in recent years, creating new drugs, as well as therapeutic regimens, increasing the number of cured patients, as well as a longer life expectancy. This might be a plausible explanation for the longer average duration of the disease in our report, since, with better HIV management, patients have greater survival that ultimately means a longer life with comorbidities (like sporotrichosis).”

Answer: Thank you for your valuable input. We had added your suggestion.

Line 291: The above text was removed from the article as it was not part of the focus.

  1. Reviewer:

I do not agree with this paragraph. Considering in Results: “Finally, speaking of the outcome, 19 of the 37 patients reported favorable evolution 232 (59.3%), 12 patients died (32.4%) and 6 patients were lost to follow-up (16.2%), so their 233 evolution is unknown (Table 1) [12–31]”.

There was a high number of patients that died (32.4%) and six patients were lost to follow-up (16.2%). The follow-up of sporotrichosis must be long because is a chronic infection with many relapses meanly in PLHIV.

Answer: Thank you for your valuable input. We had added your suggestion.

Lines 289-297: “Finally, speaking of the outcome, 24 of the 52 patients were reported with favorable evolution (46.15%), 22 patients died (42.30%) and 6 patients were lost to follow-up (16.2%), so their evolution is unknown (Table 1) [15–35]. Worth mentioning, that even though some patients were reported with favorable evolution, and others were lost before concluding their treatment, it does not mean that they could not fall back in the infection because the follow-up of sporotrichosis must be at long-term because it is a chronic infection with many relapses, especially in PLHIV. In agreement with the review by Moreira et al., we also informed that the majority of patients survived (70%)[12].”

  1. Amphotericin B is the only intravenous antifungal available to treat severe sporotrchosis and was introduced for clinical use in the 1960s. Although lipid formulations were incorporated into the therapeutic arsenal, they can still be potentially toxic and often require treatment interruption. In addition, AMB´s in vitro studies demonstrates moderate susceptibility of Sporothrix. 

Answer: Thank you for your valuable observation. The pertinent modifications were made, which are highlighted in yellow in the paper.

Lines 268 -279: “Though, it should be clarified that the chronic use of amphotericin B deoxycholate and lipid formulations can cause important nephrotoxic effects, and due to this, they are recommended mainly for the treatment of both pulmonary and meningeal sporotrichosis, and for disseminated cutaneous highly resistant to other antifungal treatments. They can also be used in cases of osteoarticular sporotrichosis where the intraarticular region is not involved. One of the great advantages of this family of drugs is its possible use during pregnancy. Another treatment that has been effective for cutaneous (fixed and lymphocutaneous) and extracutaneous (osteoarticular) forms is itraconazole. However, this drug has long-term toxic effects on the stratum corneum, causes nausea, edema, epigastrium pain, hypercholesterolemia/hypertriglyceridemia, as well as impairs liver function [39–42].”

  1. III-Conclusions:

            There are no new aspects to the subject in the conclusions to those already published      articles.

Answer: Thank you for your valuable input. We had added your suggestion.

Lines: 308 – 312:” However, paradoxically, there are cases with previous sporotrichosis that worsen at the time of initiation of treatment with retrovirals in PLHIV causing IRIS. Both IRIS and ECD sporotrichosis are the rarest and most aggressive types of sporotrichosis and therefore have a high incidence and lethality in PLHIV.”

Line 316: “and avoid the IRIS.”

Lines 319 – 321: “Nevertheless, there are cases in which it is complicated to take enough sample to make the diagnosis of Sporothrix species, as in the case of the meningitis.”

  1. Figure 1:

The quality of the Figure 1 A and C are not good.

Answer: Thank you for your valuable input. We improve the resolution of Figure 1A and 1C in line 92

Reviewer 2 Report

This is an interesting paper which mainly updates the systematic review made by Moreira et al. in 2015, with some relevant findings in the recent reports of patients with Sporotrichosis and HIV-AIDS. Some issues must be addressed:

Introduction, line 64: “This pathology can be acquired by subcutaneous traumatic inoculation (…)” Also lines 90 and 260, “pathology” or “pathologies” meaning “disease” or “diseases”. The word “pathology” refers to the study of disease, concerning nature and causes. It is mainly used related to the morphological aspects through examination of organs, tissues, etc.

Lines 77-79: “These disseminated types are usually associated with cellular immunodeficiency and inhalation infection, which explains why the population with the acquired immunodeficiency virus (HIV) gets contaminated.”  Population with the acquired immunodeficiency virus (HIV) gets contaminated in the same ways as immunocompetent individuals, but evolve more frequently with dissemination of the fungi and more severe disease than immunocompetent individuals. Please rephrase the sentence “(…) which explains why the population with the acquired immunodeficiency virus (HIV) gets contaminated.”

3.1 Epidemiology

Lines 156-158: “This could be interpreted in two ways, first that the largest number of HIV cases are from males, as reported by the epidemiology of Brazil and other affected countries such as the US and South Africa.” The second way of interpreting the predominance in males is missing. Furthermore, it is not “epidemiology of Brazil and other affected countries” but epidemiologic services of Brazil and other affected countries.

3.2 Sporotrichosis and topography

Lines 177-179: “On the other hand, in the extracutaneous presentation the most affected system was the skeletal system with 23.68 (9/38), followed by the respiratory system with 21.05% (8/38) and the central nervous system with 18.42% (7/38).” If the whole population of patients with Sporotrichosis and HIV-AIDS are 37 individuals, why there are “38” as denominators? In Table 1, extracutaneous presentation (ECD) affects 17 individuals.

4. Conclusions

Lines 250-252: “This is largely due to the fact that sporotrichosis is an emerging disease, especially in endemic regions of the world which has disseminated from zoonotic contamination by felines.” Consider rephrasing the sentence using “spread” so not to cause confusion with disseminated clinical presentation.

Lines 253-254: “Sporotrichosis infection continues to occur in a more severe and widespread manner among HIV-positive subjects with lower CD4+ counts.” Consider replacing the word “infection” for “disease”.

Author Response

Answers to Reviewer 2 concerns:

This is an interesting paper which mainly updates the systematic review made by Moreira et al. in 2015, with some relevant findings in the recent reports of patients with Sporotrichosis and HIV-AIDS.

Some issues must be addressed:

We appreciate the time and effort you have invested in the revision of our manuscript. Indeed, all your suggestions have improved the quality of our manuscript. In the main text, the additions are highlighted in yellow. We hope that we have correctly addressed all your concerns.

  1. Introduction, line 64: “This pathology can be acquired by subcutaneous traumatic inoculation (…)” Also lines 90 and 260, “pathology” or “pathologies” meaning “disease” or “diseases”. The word “pathology” refers to the study of disease, concerning nature and causes. It is mainly used related to the morphological aspects through examination of organs, tissues, etc.

Answer: Thank you for your valuable suggestion. Observations were addressed in all three lines (64, 100 and 30), pathology was changed to disease or diseases.

  1. Lines 77-79: “These disseminated types are usually associated with cellular immunodeficiency and inhalation infection, which explains why the population with the acquired immunodeficiency virus (HIV) gets contaminated.”  Population with the acquired immunodeficiency virus (HIV) gets contaminated in the same ways as immunocompetent individuals, but evolve more frequently with dissemination of the fungi and more severe disease than immunocompetent individuals. Please rephrase the sentence “(…) which explains why the population with the acquired immunodeficiency virus (HIV) gets contaminated.”

Answer: Thank you for your valuable input. We rephrase the sentence in the lines.

Lines 84 – 86: “The cutaneous disseminated or systemic forms are usually associated with cellular immunodeficiency, which explains severe sporotrichosis in PLHIV [8].”

  1. 1 Epidemiology

            Lines 156-158: “This could be interpreted in two ways, first that the largest number of HIV       cases are from males, as reported by the epidemiology of Brazil and other affected countries      such as the US and South Africa.” The second way of interpreting the predominance in males   is missing. Furthermore, it is not “epidemiology of Brazil and other affected countries” but     epidemiologic services of Brazil and other affected countries.

Answer: Thank you for your valuable input. We had added your suggestion.

Lines: 169 – 173: “This could be interpreted in two ways, first that the largest number of HIV cases are from males, as reported by the epidemiological services of Brazil and other affected countries such as the US and South Africa [36–38]. Secondly, as mentioned above, Brazil is a hyperendemic country for sporotrichosis due to the extensive zoonotic transmission in that area (cats) [9,10].“

3.2 Sporotrichosis and topography

Lines 177-179: “On the other hand, in the extracutaneous presentation the most affected system was the skeletal system with 23.68 (9/38), followed by the respiratory system with 21.05% (8/38) and the central nervous system with 18.42% (7/38).” If the whole population of patients with Sporotrichosis and HIV-AIDS are 37 individuals, why there are “38” as denominators? In Table 1, extracutaneous presentation (ECD) affects 17 individuals.

Answer: Thank you for your valuable observation. The pertinent modifications were made.

Lines 188 - 192: The most frequently observed clinical varieties were the disseminated extracutaneous (32/52), disseminated cutaneous (8/52) and disseminated (8/52) which correspond to 89.19% of the cases. In fourth place, the localized cutaneous variety was identified (2/52) in 5.41% of the patients and finally, the lymphocutaneous and cutaneous generalized varieties (1/52; 1.92%) [15–35].

  1. Conclusions

Lines 250-252: “This is largely due to the fact that sporotrichosis is an emerging disease, especially in endemic regions of the world which has disseminated from zoonotic contamination by felines.” Consider rephrasing the sentence using “spread” so not to cause confusion with disseminated clinical presentation.

Answer: Thank you for your valuable suggestion. we made the requested.

Line 305: “The prevalence of sporotrichosis coinfection per year (6.5 cases) has increased in the last 8 years with what was previously observed by Moreira et al. This is largely due to the fact that sporotrichosis is an emerging disease, especially hyperendemic regions of the world which has spread from zoonotic contamination by felines.”

  1. Lines 253-254: “Sporotrichosis infection continues to occur in a more severe and widespread manner among HIV-positive subjects with lower CD4+ counts.” Consider replacing the word “infection” for “disease”.

Answer: Thank you for your valuable suggestion. we made the requested.

Line 307: “Sporotrichosis disease continues to occur in a more severe and widespread manner among HIV-positive subjects with lower CD4+ counts.”
